# DEEP REINFORCEMENT LEARNING FOR DYNAMIC EXPECTILE RISK MEASURES: AN APPLICATION TO EQUAL RISK OPTION PRICING AND HEDGING

## ABSTRACT

Motivated by the application of equal-risk pricing and hedging of a financial derivative, where two operationally meaningful hedging portfolio policies needs to be found that minimizes coherent risk measures, we propose in this paper a novel deep reinforcement learning algorithm for solving risk-averse dynamic decision making problems. Prior to our work, such hedging problems can either only be solved based on static risk measures, leading to time-inconsistent policies, or based on dynamic programming solution schemes that are impracticable in realistic settings. Our work extends for the first time the deep deterministic policy gradient algorithm, an off-policy actor-critic reinforcement learning (ACRL) algorithm, to solving dynamic problems formulated based on time-consistent dynamic expectile risk measure. Our numerical experiments confirm that the new ACRL algorithm produces high quality solutions to equal-risk pricing and hedging problems and that its hedging strategy outperforms the strategy produced using a static risk measure when the risk is evaluated at later points of time.

## 1 INTRODUCTION

This paper considers solving risk-averse dynamic decision making problems arising from applications where risk needs to be evaluated according to risk measures that are coherent. In particular, we draw our motivation from the financial application of equal-risk pricing (ERP) and hedging (Guo & Zhu (2017)), where two dynamic hedging problems need to be solved, one for the buyer and one for the seller of a financial derivative (a.k.a option), for determining a fair transaction price that would expose both parties to the same amount of hedging risk. The need to meaningfully model each party's best hedging decision in a financial market, namely that no arbitrage is allowed, and to have a meaningful comparison between the two parties' risk exposures, namely that the risks should be measured in the same units, has led to the use of coherent risk measures for capturing both parties' hedging risks in this application (see Marzban et al. (2020)).

To this date, most solution methods proposed for solving risk-averse dynamic decision making problems under a coherent risk measure have either relied on traditional dynamic programming (DP), which suffers from the curse of dimensionality and assumes the knowledge of a stochastic model that precisely captures the dynamics of the decision environment, or on the use of a static risk measure, i.e., that disregards the temporal structure of the random variable (e.g. Marzban et al. (2020), Carbonneau & Godin (2020), and Carbonneau & Godin (2021) in the case of the ERP application). The latter raises the serious issue that the resulting policy could be time inconsistent, i.e. that the actions prescribed by the policy may be considered significantly sub-optimal once the state is visited. In an application such as ERP, this issue implies that policies obtained based on static risk measures will not be implemented in practice, raising the need to consider dynamic risk measures.

Focusing on deep reinforcement learning (DRL) methods, while there has been a large number of approaches proposed to address risk averse Markov decision processes (MDPs) using coherent risk measures, to the best of our knowledge, all of them, except for two exceptions, consider a static risk measure (see Prashanth & Ghavamzadeh (2013); Chow & Ghavamzadeh (2014); Castro et al. (2019); Singh et al. (2020); Urpí et al. (2021)) and therefore suffer from time-inconsistency. The

two exceptions consist of Tamar et al. (2015) and Huang et al. (2021) who propose actor-critic reinforcement learning (ACRL) algorithms to deal with a general dynamic law-invariant coherent risk measures. Unfortunately, the two algorithms respectively either assume that it is possible to generate samples from a perturbed version of the dynamics, or rely on training three neural networks (namely a state distribution reweighting network, a transition perturbation network, and a Lagrangean penalisation network) concurrently with the actor and critic networks. Furthermore, only Huang et al. (2021) actually implemented their method. This was done on a toy tabular problem involving 12 states and 4 actions where it produced questionable performances[1].

In this paper, we develop a new model-free ACRL algorithm for solving a time-consistent risk averse MDP under a dynamic expectile risk measure.[2] Overall, we may summarize the contribution as follows:

- Our ACRL algorithm is the first to naturally extend the popular model-free deep deterministic policy gradient algorithm (DDPG) (see Lillicrap et al. (2015)) to a risk averse setting where a time consistent coherent risk measure is used. Unlike the ACRL proposed in Huang et al. (2021), which employs five neural networks, our algorithm will only require an actor and a critic network. While our policy network will be trained following a stochastic gradient procedure similar to Silver et al. (2014), we are the first to leverage the elicitability property of expectile risk measures to propose a procedure for training the "risk-to-go" deep Q-network that is also based on stochastic gradient descent.

- Our ACRL is the first model-free DRL-based algorithm capable of identifying optimal risk averse option hedging strategies that are time-consistent with respect to a dynamic coherent risk measure, and of computing their associated equal risk prices. A side benefit of time-consistency will be that after training for an option with a given maturity, one obtains equal risk prices and hedging strategies for any other shorter maturities. While our algorithm certainly has a broader set of applications, we believe that ERP constitutes an original and fertile application in which to develop and test new risk averse DRL methods.

- We evaluate the training efficiency and the quality of solution, in terms of quality of option hedging strategies and of estimated equal risk prices, obtained using our ACRL algorithm on a synthetic multi-asset geometric Brownian motion market model. These experiments constitute the first real application of a risk averse DRL algorithm that employs a dynamic coherent risk measure.

The rest of this paper is organized as follows. Section 2 introduces equal risk pricing and its associated DP equations. Section 3 proposes the new ACRL algorithm for general finite horizon risk averse MDP with dynamic expectile measures. Finally, Section 4 presents and discusses our numerical experiments. We note that a reader only interested in the ACRL algorithm can skip right ahead to Section 3.

## 2 APPLICATION: EQUAL RISK PRICING AND HEDGING UNDER DYNAMIC EXPECTILE RISK MEASURES

As described in Marzban et al. (2020), the problem of ERP can be formalized as follows. Consider a frictionless market, i.e. no transaction cost, tax, etc, that contains $m$ risky assets, and a risk-free bank account with zero interest rate. Let $\mathbf{S}_t : \Omega \to \mathbb{R}^m$ denote the values of the risky assets adapted to a filtered probability space $(\Omega, \mathcal{F}, \mathbb{F} := \{\mathcal{F}_t\}_{t=0}^T, \mathbb{P})$, i.e. each $\mathbf{S}_t$ is $\mathcal{F}_t$ measurable. It is assumed that $\mathbf{S}_t$ is a locally bounded real-valued semi-martingale process and that the set of equivalent local martingale measures is non-empty (i.e. no arbitrage opportunity). The set of all admissible self-financing hedging strategies with the initial capital $p_0 \in \mathbb{R}$ is shown by $\mathcal{X}(p_0)$:

$$\mathcal{X}(p_0) = \left\{ X : \Omega \to \mathbb{R}^T \,\middle|\, \exists \{\boldsymbol{\xi}_t\}_{t=0}^{T-1}, \quad X_t = p_0 + \sum_{t'=0}^{t-1} \boldsymbol{\xi}_{t'}^\top \Delta \mathbf{S}_{t'+1}, \quad \forall t = 1, \ldots, T \right\},$$

---

[1]At the time of writing this paper, the risk averse implementation of this algorithm reported in Huang et al. (2021) is unable to recommend an optimal policy in a deterministic setting, while the risk neutral implementation produces policies that are outperformed by risk averse ones in a stochastic setting.

[2]Our ACRL algorithm exploits the elicitabilty property of expectile risk measures, which is the only elicitable coherent risk measure.

where $\Delta\mathbf{S}_{t+1} := \mathbf{S}_{t+1} - \mathbf{S}_t$, the hedging strategy $\boldsymbol{\xi}_t \in \mathbb{R}^m$ is a vector of random variables adapted to the filtration $\mathbb{F}$ and captures the number of shares of each risky asset held in the portfolio during the period $[t,\ t+1]$, and $X_t$ is the accumulated wealth.

Let $F(\{\mathbf{S}_t\}_{t=1}^T)$ denote the payoff of a derivative. Throughout this paper, we assume $F(\{\mathbf{S}_t\}_{t=1}^T)$ admits the formulation of $F(\mathbf{S}_T, \mathbf{Y}_T)$ where $\mathbf{Y}_t$ is an auxiliary fixed-dimensional stochastic process that is $\mathcal{F}_t$-measurable. This class of payoff functions is common in the literature, (see for example Bertsimas et al. (2001) and Marzban et al. (2020)). The problem of ERP is defined based on the following two hedging problems that seek to minimize the risk of hedging strategies, one is for the writer and the other is for the buyer of the derivative:

$$\text{(Writer)} \qquad \varrho^w(p_0) = \inf_{X \in \mathcal{X}(p_0)} \rho^w(F(\mathbf{S}_T, \mathbf{Y}_T) - X_T) \tag{1}$$

$$\text{(Buyer)} \qquad \varrho^b(p_0) = \inf_{X \in \mathcal{X}(-p_0)} \rho^b(-F(\mathbf{S}_T, \mathbf{Y}_T) - X_T), \tag{2}$$

where $\rho^w$ and $\rho^b$ are two risk measures that capture respectively the writer and the buyer's risk aversion. In words, equation (1) describes a writer that is receiving $p_0$ as the initial payment and implements an optimal hedging strategy for the liability $F(\mathbf{S}_T, \mathbf{Y}_T)$. On the other hand, in (2) the buyer is assumed to borrow $p_0$ in order to pay for the option and then to manage a portfolio that will minimize the risks associated to his final wealth. With equations (1) and (2), ERP defines a fair price $p_0^*$ as the value of an initial capital that leads to the same risk exposure to both parties, i.e. $\varrho^w(p_0^*) = \varrho^b(p_0^*)$.

In particular, based on Proposition 3.1 and the examples presented in section 3.3 of Marzban et al. (2020), together with the fact that both $\rho^w$ and $\rho^b$ are dynamic recursive law invariant risk measures, a Markovian assumption allows us to conclude that the ERP can be calculated using two sets of dynamic programming equations.

**Assumption 2.1.** *[Markov property] There exists a sufficient statistic process $\psi_t$ adapted to $\mathbb{F}$ such that $\{(\mathbf{S}_t, \mathbf{Y}_t, \psi_t)\}_{t=0}^T$ is a Markov process relative to the filtration $\mathbb{F}$. Namely, $\mathbb{P}((\mathbf{S}_{t+s}, \mathbf{Y}_{t+s}, \psi_{t+s}) \in \mathcal{A}|\mathcal{F}_t) = \mathbb{P}((\mathbf{S}_{t+s}, \mathbf{Y}_{t+s}, \psi_{t+s}) \in \mathcal{A}|\mathbf{S}_t, \mathbf{Y}_t, \psi_t)$ for all t, for all $s \geq 0$, and all sets $\mathcal{A}$.*

Specifically, on the writer side, we can define $V_T^w(\mathbf{S}_T, \mathbf{Y}_T, \psi_T) := F(\mathbf{S}_T, \mathbf{Y}_T)$, and recursively

$$V_t^w(\mathbf{S}_t, \mathbf{Y}_t, \psi_t) := \inf_{\boldsymbol{\xi}_t} \bar{\rho}(-\boldsymbol{\xi}_t^\top \Delta\mathbf{S}_{t+1} + V_{t+1}^w(S_t + \Delta\mathbf{S}_{t+1}, \mathbf{Y}_t + \Delta Y_{t+1}, \psi_{t+1})|\mathbf{S}_t, \mathbf{Y}_t, \psi_t),$$

where $\bar{\rho}(\cdot|\mathbf{S}_t, \mathbf{Y}_t, \psi_t)$ is a law invariant risk measure that uses $\mathbb{P}(\cdot|\mathbf{S}_t, \mathbf{Y}_t, \psi_t)$. This leads to considering $\varrho^w(0) = V_0^w(\mathbf{S}_0, \mathbf{Y}_0, \psi_0)$. On the other hand, for the buyer we similarly define: $V_T^b(\mathbf{S}_T, \mathbf{Y}_T, \psi_T) := -F(\mathbf{S}_T, \mathbf{Y}_T)$ and

$$V_t^b(\mathbf{S}_t, \mathbf{Y}_t, \psi_t) := \inf_{\boldsymbol{\xi}_t} \bar{\rho}(-\boldsymbol{\xi}_t^\top \Delta\mathbf{S}_{t+1} + V_{t+1}^b(\mathbf{S}_t + \Delta\mathbf{S}_{t+1}, \mathbf{Y}_t + \Delta\mathbf{Y}_{t+1}, \psi_{t+1})|\mathbf{S}_t, \mathbf{Y}_t, \psi_t),$$

with $\varrho^b(0) = V_0^b(\mathbf{S}_0, \mathbf{Y}_0, \psi_0)$. The following lemma summarizes how DP can be used to compute the ERP.

**Lemma 2.1** (Marzban et al. (2020)). *Under Assumption 2.1, the ERP that employs dynamic expectile risk measure can be computed as: $p_0^* = (V_0^w(\mathbf{S}_0, \mathbf{Y}_0, \psi_0) - V_0^b(\mathbf{S}_0, \mathbf{Y}_0, \psi_0))/2$.*

In what follows, we will further assume that the risk measure is a dynamic expectile risk measure.

**Definition 2.1.** *A dynamic expectile risk measure takes the form: $\rho(X) := \bar{\rho}_0(\bar{\rho}_1(\ldots\bar{\rho}_{T-1}(X)))$ where each $\bar{\rho}(\cdot)$ is an expectile risk measure that employs the conditional distribution based on $\mathcal{F}_t$. Namely, $\bar{\rho}_t(X_{t+1}) := \arg\min_q \tau\mathbb{E}\left[(q - X_{t+1})_+^2|\mathcal{F}_t\right] + (1 - \tau)\mathbb{E}\left[(q - X_{t+1})_-^2|\mathcal{F}_t\right]$ where $X_{t+1}$ is a random liability measureable on $\mathcal{F}_{t+1}$.*

Like conditional value at risk, the expectile measure (see Bellini & Bignozzi (2015)) covers the range of risk attitudes from risk neutrality, when $\tau = 1/2$, to worst-case risk, when $\tau \to 1$.

# 3 A NOVEL ACTOR-CRITIC ALGORITHM FOR RISK AVERSE MDP UNDER A DYNAMIC EXPECTILE RISK MEASURE

With the dynamic programming equations in hand, it now becomes apparent that each option hedging problem in ERP can be formulated as a finite horizon Markov Decision Process (MDP) described

with $(\mathcal{S}, \mathcal{A}, r, P)$. In this regard, the agent (i.e. the writer or buyer) interacts with a stochastic environment by taking an action $a_t \equiv \boldsymbol{\xi}_t \in [-1, 1]^m$ after observing the state $s_t \in \mathcal{S}$, which includes $\mathbf{S}_t$, $\mathbf{Y}_t$, and $\psi_t$. Note that to simplify exposition, in this section we drop the reference to the specific identity (i.e. $w$ or $b$) of the agent in our notation. The action taken at each time $t$ results in the immediate stochastic reward that takes the shape of the immediate hedging portfolio return, i.e. $r_t(s_t, a_t, s_{t+1}) := \boldsymbol{\xi}_t^\top \Delta \mathbf{S}_{t+1}$ when $t < T$ and otherwise of the option liability/payout $r_T(s_T, a_T, s_{T+1}) := F(\mathbf{S}_T, \mathbf{Y}_T)(1 - 2 \cdot \mathbf{1}\{\text{agent=writer}\})$, which is insensitive to $s_{T+1}$. Finally, the Markovian exogeneous dynamics described in Assumption 2.1 are modeled using $P$ as $P(s_{t+1}|s_t, a_t) = \mathbb{P}(\mathbf{S}_{t+1}, \mathbf{Y}_{t+1}, \psi_{t+1}|\mathbf{S}_t, \mathbf{Y}_t, \psi_t)$. Overall, each of the two dynamic derivative hedging problems presented in Section 2 reduce to a version of the following general risk averse reinforcement learning problem:

$$\varrho(0) = V_0(\mathbf{S}_0, \mathbf{Y}_0, \psi_0) = \min_\pi Q_0^\pi(\bar{s}_0, \pi_0(\bar{s}_0)) \,,$$

where $\bar{s}_0 := (\mathbf{S}_0, \mathbf{Y}_0, \psi_0)$ is the initial state in which the option is priced while $Q_t^\pi(s_t, a_t) := \bar{\rho}(-r_t(s_t, a_t, s_{t+1}) + Q_{t+1}^\pi(s_{t+1}, \pi_{t+1}(s_{t+1}))|s_t)$, $Q_T^\pi(s_T, a_T) := -r_T(s_T, a_T, s_T)$, and where $\bar{\rho}$ is an expectile risk measure, i.e. $\bar{\rho}(X) := \arg\min_q \mathbb{E}\left[\tau(q - X)_+^2 + (1 - \tau)(q - X)_-^2\right]$. Equipped with these definitions, we can now motivate our proposed extension of the model-free off-policy deterministic ACRL algorithm to the general finite horizon risk-averse MDP setting. In doing so, we start with a proposition (see Appendix A.1 for a proof) that will provide the motivation for a stochastic gradient scheme to optimize the actor network, while the optimization of the critic network will follow from the elicitability property of the expectile risk measure.

**Proposition 3.1.** *Let $\bar{\pi}$ be an arbitrary reference policy and $\mu$ an arbitrary distribution over the initial state, such that there is a strictly positive probability of reaching all of $\mathcal{S}$ for all $t \geq 1$.[3] For any $\pi^*$ that satisfies*

$$\pi^* \in \arg\min_\pi \mathbb{E}_{\substack{\tilde{t} \sim \{0,\dots,T\}, s_0 \sim \mu \\ s_{t+1} \sim P(\cdot|s_t, \bar{\pi}_t(s_t))}} \left[Q_{\tilde{t}}^\pi(s_{\tilde{t}}, \pi_{\tilde{t}}(s_{\tilde{t}}))\right] \tag{3}$$

*where $\tilde{t}$ is uniformly drawn, we necessarily have that $\pi^* \in \arg\min Q_0^\pi(\bar{s}_0, \pi_0(\bar{s}_0))$ hence $\varrho(0) = Q_0^{\pi^*}(\bar{s}_0, \pi_0^*(\bar{s}_0))$.*

In the context of a deep reinforcement learning approach, we can employ a procedure based on off-policy deterministic policy gradient (Silver et al., 2014) to optimize (3). Specifically, given a policy network $\pi^\theta$, we wish to optimize:

$$\min_\theta \mathbb{E}_{\substack{\tilde{t} \sim \{0,\dots,T-1\} \\ s_{t+1} \sim P(\cdot|s_t, \bar{\pi}_t(s_t))}} \left[Q_{\tilde{t}}^{\pi^\theta}(s_{\tilde{t}}, \pi_{\tilde{t}}^\theta(s_{\tilde{t}}))\right] \,,$$

using a stochastic gradient algorithm. In doing so, we rely on the fact that:

$$\nabla_\theta \mathbb{E}_{\substack{\tilde{t} \sim \{0,\dots,T-1\} \\ s_{t+1} \sim P(\cdot|s_t, \bar{\pi}_t(s_t))}} \left[Q_{\tilde{t}}^{\pi^\theta}(s_{\tilde{t}}, \pi^\theta(s_{\tilde{t}}))\right]$$

$$= \mathbb{E}_{\substack{\tilde{t} \sim \{0,\dots,T-1\} \\ s_{t+1} \sim P(\cdot|s_t, \bar{\pi}_t(s_t))}} \left[\nabla_\theta Q_{\tilde{t}}^{\pi^\theta}(s_{\tilde{t}}, a)\Big|_{a=\pi_{\tilde{t}}^\theta(s_{\tilde{t}})} + \nabla_a Q_{\tilde{t}}^{\pi^\theta}(s_{\tilde{t}}, a)\nabla_\theta \pi_{\tilde{t}}^\theta(s_{\tilde{t}})\Big|_{a=\pi_{\tilde{t}}^\theta(s_{\tilde{t}})}\right]$$

$$\approx \mathbb{E}_{\substack{\tilde{t} \sim \{0,\dots,T-1\} \\ s_{t+1} \sim P(\cdot|s_t, \bar{\pi}_t(s_t))}} \left[\nabla_a Q_{\tilde{t}}^{\pi^\theta}(s_{\tilde{t}}, a)\nabla_\theta \pi_{\tilde{t}}^\theta(s_{\tilde{t}})\Big|_{a=\pi_{\tilde{t}}^\theta(s_{\tilde{t}})}\right] \,.$$

Note that in the above equation, we have dropped the term that depends on $\nabla_\theta Q_{\tilde{t}}^{\pi^\theta}$ as is commonly done in off-policy deterministic gradient methods and usually motivated by a result of Degris et al. (2012), who argue that this approximation preserves the set of local optima in a risk neutral setting, i.e. $\bar{\rho}(\cdot) := \mathbb{E}[\cdot]$. While we do consider as an important subject of future research to extend this motivation to more general risk measures, our numerical experiments (see Section 4.3) will confirm empirically that the quality of this approximation permits the identification of nearly optimal hedging policies.

---

[3] In our option hedging problem, given that $s_t$ is entirely exogenous, the distribution of $s_{t+1}$ is unaffected by $\bar{\pi}$, which can therefore be chosen arbitrarily. Moreover, $\mu$ can put all the mass on $\bar{s}_0$.

Given that we do not have access to an exact expression for $Q_{\tilde{t}}^{\pi^\theta}(s_{\tilde{t}}, a)$, this operator needs to be estimated directly from the training data. Exploiting the fact that $\rho$ is a utility-based shortfall risk measure, we get that:

$$Q_t^\pi(s_t, a_t) \in \arg\min_q \mathbb{E}_{s_{t+1} \sim P(\cdot|s_t, a_t)}[\ell(q + r_t(s_t, a_t, s_{t+1}) - Q_{t+1}^\pi(s_{t+1}, \pi_{t+1}(s_{t+1})))]$$

where $\ell(y) := (\tau \mathbf{1}\{y > 0\} - (1-\tau)\mathbf{1}\{y \le 0\})y^2$ is the score function associated to the $\tau$-expectile risk measure. As explained in Theorem 3.2 of Shen et al. (2014), in a tabular MDP environment one can apply the following stochastic gradient step:

$$\hat{Q}_t(s_t, a_t) \leftarrow \hat{Q}_t(s_t, a_t) - \alpha \partial\ell(\hat{Q}_t(s_t, a_t) + r_t(s_t, a_t, s_{t+1}) - \hat{Q}_{t+1}(s_{t+1}, \pi_{t+1}(s_{t+1})),$$

where $\partial\ell(y) := 2(\tau\max(0, y) - (1-\tau)\max(0, -y))$ is the derivative of $\ell(y)$, within a properly designed Q-learning algorithm and have the guarantee that $\hat{Q}_t(s_t, a_t)$ will almost surely converge to $Q_t^\pi(s_t, a_t)$ for all $t$, $s_t$, and $a_t$.

In the non-tabular setting, we replace $\hat{Q}_t^\pi(s_t, a_t)$ with two estimators: i.e. the "main" network $Q_t^\pi(s_t, a_t|\theta^Q)$ for the immediate conditional risk and the "target" network $Q_t^\pi(s_t, a_t|\theta^{Q'})$ for the next period's conditional risk. The procedure consists in iterating between a step that attempts to make the main network $Q_t^\pi(s_t, a_t|\theta^Q)$ a good estimator of $\rho(-r(s_t, a_t, s_{t+1}) + Q_{t+1}^\pi(s_{t+1}, a_{t+1}|\theta^{Q'}))$ and a step that replaces the target network $Q_t^\pi(s_t, a_t|\theta^{Q'})$ with a network more similar to the main one $Q_t^\pi(s_t, a_t|\theta^Q)$. The former is achieved, similarly as with the policy network, by searching for the optimal $\theta^Q$ according to:

$$\min_{\theta^Q} \mathbb{E}_{\substack{\tilde{t} \sim \{0,\ldots,T-1\} \\ s_{t+1} \sim P(\cdot|s_t, \bar{\pi}_t(s_t))}} [\ell(Q_{\tilde{t}}^\pi(s_{\tilde{t}}, \bar{\pi}_{\tilde{t}}(s_{\tilde{t}})|\theta^Q) + r_t(s_{\tilde{t}}, \bar{\pi}_{\tilde{t}}(s_{\tilde{t}}), s_{\tilde{t}+1}) - Q_{\tilde{t}+1}^\pi(s_{\tilde{t}+1}, \pi_{\tilde{t}+1}(s_{\tilde{t}+1})|\theta^{Q'}))],$$

which suggests a stochastic gradient update of the form $\theta^Q \leftarrow \theta^Q - \alpha\Delta$, where $\Delta$ is

$$\partial\ell(Q_{\tilde{t}}^\pi(s_{\tilde{t}}, \bar{\pi}_{\tilde{t}}(s_{\tilde{t}})|\theta^Q) + r_t(s_{\tilde{t}}, \bar{\pi}_{\tilde{t}}(s_{\tilde{t}}), s_{\tilde{t}+1}) - Q_{\tilde{t}+1}^\pi(s_{\tilde{t}+1}, \pi_{\tilde{t}+1}(s_{\tilde{t}+1})|\theta^{Q'}))\nabla_{\theta^Q}Q_{\tilde{t}}^\pi(s_{\tilde{t}}, \bar{\pi}_{\tilde{t}}(s_{\tilde{t}})|\theta^Q).$$

These two types of updates are integrated in our proposed expectile-based actor-critic deep RL (a.k.a. ACRL) algorithm. A first version, Algorithm 1, is designed for a simulation-based environment. One may note that in each episode, the reference policy $\bar{\pi}_t$ is updated to be a perturbed version of the main policy network in order to focus the accuracy of the main critic network's value and derivatives on actions that are more likely to be produced by the main policy network. We also choose to update the target networks using convex combinations operations as is done in Lillicrap et al. (2015) in order to improve stability of learning. A second more general version of ACRL, which mimics the original DDPG, by generating minibatches using a replay buffer can also be found in Appendix A.2.

We finally note that in our problem, $P(s_{t+1}|s_t, a_t) = P(s_{t+1}|s_t, a_t') = \mathbb{P}(\mathbf{S}_{t+1}, \mathbf{Y}_{t+1}, \psi_{t+1}|\mathbf{S}_t, \mathbf{Y}_t, \psi_t)$, meaning that the action is not affecting the distribution of state in the next period. This is a direct consequence of using a translation invariant risk measure, which eliminates the need to keep track of the accumulated wealth in the set of state variables as explained in Marzban et al. (2020) and allows the reward function to provide an immediate signal regarding the quality of implemented actions. In the context of our deep reinforcement learning approach, we observed that convergence speed is significantly improved in training due to this property (see Figure 4 in Appendix).

## 4 EXPERIMENTAL RESULTS

In this section we provide two different sets of experiments that are run over one vanilla and one basket option. We will compare both algorithmic efficiency and quality, in terms of pricing and hedging strategies, of the dynamic risk model (DRM), which employs a dynamic expectile risk measure and is solved using our new ACRL algorithm, and the static risk model (SRM), which employs a static expectile measure and is solved using an AORL algorithm similar to Carbonneau & Godin (2021). All experiments are done using simulated price processes of five risky assets: AAPL, AMZN, FB, JPM, and GOOGL. The price paths are simulated using correlated Brownian motions considering the empirical mean, variance, and the correlation matrix of five reference stocks (AAPL, AMZN, FB, KPM, and GOOGL) over the period that spans from January 2019 to January 2021. In

---

**Algorithm 1:** The actor-critic RL algorithm for the dynamic recursive expectile option hedging problem (ACRL)

---

Randomly initialize the main actor and critic networks' parameters $\theta^\pi$ and $\theta^Q$;

Initialize the target actor, $\theta^{\pi'} \leftarrow \theta^\pi$, and critic, $\theta^{Q'} \leftarrow \theta^Q$, networks;

**for** $j = 1 : \#Episodes$ **do**

    Randomly select $t \in \{0, 1, ..., T-1\}$;

    Sample a minibatch of N triplets $\{(s_t^i, a_t^i, s_{t+1}^i)\}_{i=1}^N$ from $P(\cdot|s_t, \bar{\pi}_t(s_t))$, where $\bar{\pi}_t(s_t) := \pi_t(s_t|\theta^\pi) + \mathcal{N}(0, \sigma)$;

    Set the realized losses $y_t^i := -r_t(s_t^i, a_t^i, s_{t+1}^i) + Q_{t+1}(s_{t+1}^i, \pi_{t+1}(s_{t+1}^i|\theta^{\pi'})|\theta^{Q'})$;

    Update the main critic network:

$$\theta^Q \leftarrow \theta^Q - \alpha \frac{1}{N} \sum_{i=1}^N \partial\ell(Q_t(s_t^i, a_t^i|\theta^Q) - y_t^j) \nabla_{\theta^Q} Q_t(s_t^i, a_t^i|\theta^Q);$$

    Update the main actor network:

$$\theta^\pi \leftarrow \theta^\pi - \alpha \frac{1}{N} \sum_{i=1}^N \nabla_a Q_t(s_t^i, a|\theta^Q)|_{a=\pi_t(s_t^i|\theta^\pi)} \nabla_{\theta^\pi} \pi_t(s_t^i|\theta^\pi);$$

    Update the target networks:

$$\theta^{Q'} \leftarrow \alpha\theta^Q + (1-\alpha)\theta^{Q'}, \qquad \theta^{\pi'} \leftarrow \alpha\theta^\pi + (1-\alpha)\theta^{\pi'}; \qquad (4)$$

**end**

---

both experiments, the maturity of the option will be one year and the hedging portfolios will be rebalanced on a monthly basis. Table 3 in the appendix provides the descriptive statistics of our underlying hidden stochastic process.

In what follows, we first explain the architectures of our ACRL model. Then, the training procedure of the networks under the dynamic risk measurement is elaborated. Finally, the main numerical results of the paper are presented for pricing and hedging a vanilla, where the precision of our approach will be empirically demonstrated, and a basket option. All codes are available at https://anonymous.4open.science/r/ERP-Dynamic-Expectile-RM-4BEA.

### 4.1 ACTOR AND CRITIC NETWORK ARCHITECTURE

Our implementation of the ACRL algorithm involves two simple networks presented in Figure 1. Since the underlying assets follow a Brownian motion, the actor and critic networks can define the input state as the logarithm of the cumulative returns of each asset and the time remaining to maturity (i.e. dimension = $m + 1$). The actor network is composed of three fully connected layers where the number of neurons are considered to be $k = 32$ in the first two layers and then maps back to the number of assets in the last layer to generate the investment policy accordingly for each asset. The activation functions in our networks are considered to be *tanh* functions. In the last layer, this implies that the actions will lie in $[-1, 1]^m$. The critic network only concatenates the $m$ dimensional action information vector after its third layer. In the case of SRM, only the actor network is used.

### 4.2 TRAINING PROCEDURE AND LEARNING CURVES

Recall that in an SRM setting, overfitting of any DRL algorithm can be controlled by measuring the performance of the trained policy on a validation data set using an empirical estimate of the risk-averse objective as validation score. Unfortunately, this is no longer possible in the case of DRMs since the risk measure relies on conditional risk measurements of the trajectories produced by our policy. In theory, estimates of such conditional measurements could be obtained by training a new critic network using the validation set (while maintaining the policy fixed to the trained one). In practice, this is highly computationally demanding to perform in the training stage and raises a new

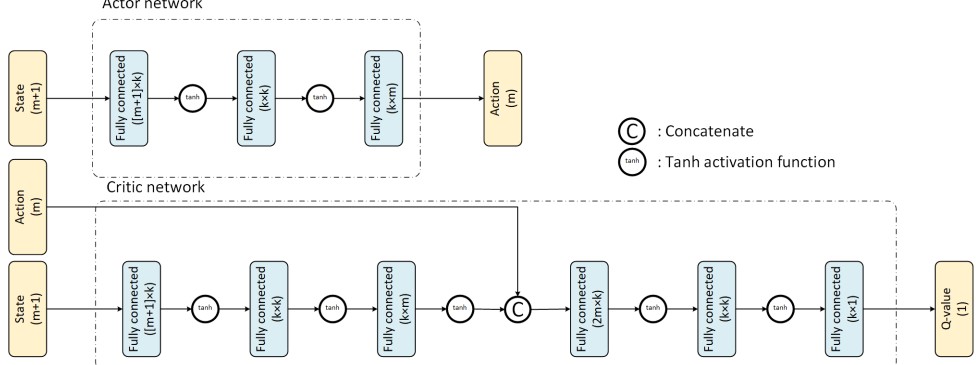

Figure 1: The architecture of the actor and critic networks in ACRL algorithm.

issue of how to control overfitting of the validation score estimate. Our solution for this problem is to rely on using static risk measures as validation score, namely a set of static expectiles at risk levels that are larger or equal to the risk level of the DRM. Figures 2 and 3 in the appendix show examples of learning curves for the validation performance of DRM and SRM approaches on vanilla and basket options at a risk level of $\tau = 90\%$, with a maturity $T = 12$. SRM appeared to have a faster rate of convergence than DRM, due to its simpler architecture. Being a time-inconsistent model, SRM must however be retrained whenever the maturity of the option is modified. When comparing convergence rates between vanilla and basket options, we observed similar behavior, which indicates that the training time might not be very sensitive to the number of assets, thus suffering less from the curse of dimensionality. We finally note that both our training and validation sets included 1000 trajectories from the underlying geometric Brownian motion process, implying that the procedure can be applied in settings where only historical data is available.

### 4.3 VANILLA OPTION HEDGING AND PRICING

In our first set of experiments, we consider pricing and hedging an at-the-money vanilla call option on AAPL. In this setting, it is possible to obtain (approximately) optimal solutions by dynamic programming via discretization of the state space. The initial price of AAPL is set to 78.81 and options with time to maturity ranging from one month to one year are considered. Both DRM and SRM are trained using a one year maturity/horizon.

With the trained DRM and SRM policy networks, we can evaluate the writer and the buyer's (out-of-sample) risk exposure over a pre-specified time horizon so as to calculate the corresponding ERP. We consider the following three metrics for measuring the realized risk under different hedging policy and explain the methods used for calculating the metrics:

- *Out-of-sample static expectile risk*: Given a trained policy, use the test data to calculate the static expectile risk. This is the metric that should be minimized by the SRM.

- *RL based out-of-sample dynamic expectile risk estimation*: Given the trained policy, use the test data to only train a critic network using ACRL to produce an estimate of out-of-sample dynamic expectile risk. This is an estimate of the metric minimized by the DRM.

- *DP based out-of-sample dynamic expectile risk estimation*: Given a trained policy, evaluate the "true" dynamic expectile risk by solving the dynamic programming equations using a high precision discretization of the states, actions, and transitions.[4] This serves as the true metric minimized by the DRM.

We note that our RL based estimate of out-of-sample dynamic risk is a novel approach, which tackles the important challenge of policy evaluation in RL with dynamic risk measures.

Table 1 summarizes the evaluations of out-of-sample dynamic risk for DRM policies trained for 1 year maturity at risk level $\tau = 90\%$ then applied to options of different maturities ranging from 12

---

[4]Note that this metric is available neither for the case of basket option nor in a data-driven environment.

Table 1: The out-of-sample dynamic and static 90%-expectile risk imposed to the two sides of vanilla at-the-money call options over AAPL.

| Policy | Est.[†] | \multicolumn{9}{c}{Time to maturity} |
| --- | --- | --- | --- | --- | --- | --- | --- | --- | --- | --- | --- |
| | | 12 | 11 | 10 | 9 | 8 | $\cdots$ | 4 | 3 | 2 | 1 |
| \multicolumn{12}{c}{Dynamic 90%-expectile risk} |
| Writer's DRM | RL | 0.77 | 0.73 | 0.69 | 0.65 | 0.62 | $\cdots$ | 0.45 | 0.38 | 0.29 | 0.23 |
| | DP | 0.75 | 0.71 | 0.68 | 0.65 | 0.61 | $\cdots$ | 0.43 | 0.38 | 0.31 | 0.23 |
| Buyer's DRM | RL | -0.22 | -0.21 | -0.20 | -0.19 | -0.18 | $\cdots$ | -0.11 | -0.09 | -0.07 | -0.05 |
| | DP | -0.23 | -0.22 | -0.21 | -0.20 | -0.18 | $\cdots$ | -0.12 | -0.11 | -0.08 | -0.06 |
| \multicolumn{12}{c}{Static 90%-expectile risk} |
| Writer's SRM | ED | 0.55 | 0.54 | 0.54 | 0.53 | 0.53 | $\cdots$ | 0.48 | 0.46 | 0.41 | 0.31 |
| Writer's DRM | ED | 0.56 | 0.54 | 0.52 | 0.50 | 0.47 | $\cdots$ | 0.36 | 0.33 | 0.29 | 0.24 |
| Buyer's SRM | ED | -0.35 | -0.33 | -0.30 | -0.27 | -0.23 | $\cdots$ | -0.09 | -0.07 | -0.07 | -0.06 |
| Buyer's SRM | ED | -0.36 | -0.34 | -0.32 | -0.30 | -0.28 | $\cdots$ | -0.18 | -0.14 | -0.11 | -0.06 |
| \multicolumn{12}{c}{Equal risk prices with DRM} |
| True ERP | | 0.49 | 0.47 | 0.45 | 0.42 | 0.40 | $\cdots$ | 0.28 | 0.24 | 0.19 | 0.14 |
| DRM | RL | 0.50 | 0.47 | 0.45 | 0.42 | 0.40 | $\cdots$ | 0.28 | 0.24 | 0.18 | 0.14 |
| SRM | RL | 0.49 | 0.46 | 0.44 | 0.43 | 0.40 | $\cdots$ | 0.30 | 0.27 | 0.24 | 0.22 |

[†] Estimation (Est.) is either made based on reinforcement learning (RL), discretized dynamic programming (DP), or the empirical distribution (ED).

months to 1 months. One can observe that the risk of the writer decreases monotonically for options of shorter maturities, whereas the risk of the buyer increases monotonically. This is consistent with the fact that there is less uncertainty for a shorter hedging horizon, which favors the writer's risk exposure more than the buyer's when considering an at-the-money option. This also provides the evidence that the DRM policies, albeit only trained based on the longest time to maturity, i.e. one year, can be well applied to hedge options with shorter time to maturity and be used to draw consistent conclusion. Another important observation one can make is that the RL based out-of-sample dynamic risk estimate is generally very close to the DP based estimate across all conditions.

Table 1 also reports the out-of-sample static risk for both SRM policies and DRM policies. The results are interesting and perhaps surprising. First, the DRM policies outperform SRM policies in terms of static risk exposure for short maturities, even though they were trained using a different risk measure. Second, unlike with DRM, we observed at other risk levels (see Figure 6(e) and (f) in Appendix) that the static risk of SRM policies for the seller (resp. buyer) can increase (resp. decrease) when hedging an option with shorter maturity. The possibility that a seller's policy may actually increase risk when applied to an option with shorter maturity is clearly problematic here as it is inconsistent with the fact that there is less uncertainty (and lower expected value) regarding the payout of such options. Both observed phenomenon are consequences of the fact that SRM violates the time consistency property. We suspect that the possibility that SRM policies may not account properly for risk aversion at some future time point or for other range of option maturities should seriously hinder their use in practice.

Finally, Table 1 reports the equal risk prices calculated based on RL based out-of-sample dynamic risk estimate and based on the discretized DP (referred as True ERP).[5] One first confirms that the RL based estimate is of high quality, with a maximum approximation error of 0.01 over all maturities. Moreover, we can see that the prices for the SRM polices are generally higher than the prices for the DRM polices, perhaps due to the fact that it is the writer that benefits most from the improved DRM policy than the buyer, as he is more exposed to tail risks in this transaction. We further refer the reader to section B.2 of the appendix for additional results regarding the performance of SRM and DRM in this vanilla option setting.

## 4.4 BASKET OPTION HEDGING AND PRICING

In our second set of experiments, we extend the application of ERP pricing framework to the case of basket options where traditionnal DP solution schemes are not computationally tractable. In

---

[5]Note that in a real data-driven setting, the ERP could either be estimated using the in-sample trained critic network, or by calculating our RL based estimate using some freshly reserved data to reduce statistical biases.

particular, we consider an at-the-money basket option with the strike price of 753$ on five underlying assets: AAPL, AMZN, FB, JPM, and GOOGL, where the option payoff is determined by the average price of the underlyings. In this section, dynamic risk is only estimated using the RL based estimator defined in Section 4.3, given that exact DP resolution has become intractable.

Table 2: The out-of-sample dynamic and static 90%-expectile risk imposed to the two sides of basket at-the-money call options. Associated ERPs under the DRM are also compared.

| Policy | Est.[†] | Time to maturity | | | | | | | | | |
|---|---|---|---|---|---|---|---|---|---|---|---|
| | | 12 | 11 | 10 | 9 | 8 | ⋯ | 4 | 3 | 2 | 1 |
| Dynamic 90%-expectile risk | | | | | | | | | | | |
| Writer's DRM | RL | 3.92 | 3.62 | 3.38 | 3.15 | 2.95 | ⋯ | 2.00 | 1.70 | 1.39 | 1.10 |
| Buyer's DRM | RL | -0.48 | -0.49 | -0.51 | -0.52 | -0.50 | ⋯ | -0.47 | -0.37 | -0.33 | -0.29 |
| Static 90%-expectile risk | | | | | | | | | | | |
| Writer's SRM | ED | 2.43 | 2.36 | 2.28 | 2.16 | 2.08 | ⋯ | 1.61 | 1.45 | 1.26 | 0.94 |
| Writer's DRM | ED | 2.38 | 2.28 | 2.18 | 2.06 | 1.96 | ⋯ | 1.51 | 1.39 | 1.20 | 0.92 |
| Buyer's SRM | ED | -1.31 | -1.24 | -1.15 | -1.01 | -0.94 | ⋯ | -0.56 | -0.48 | -0.36 | -0.22 |
| Buyer's SRM | ED | -1.39 | -1.32 | -1.24 | -1.13 | -1.07 | ⋯ | -0.66 | -0.56 | -0.40 | -0.23 |
| Equal risk prices with DRM | | | | | | | | | | | |
| DRM | RL | 2.20 | 2.06 | 1.95 | 1.84 | 1.73 | ⋯ | 1.24 | 1.04 | 0.86 | 0.70 |
| SRM | RL | 2.23 | 2.10 | 2.01 | 1.91 | 1.79 | ⋯ | 1.21 | 1.03 | 0.92 | 0.82 |

[†] Estimation (Est.) is either made based on reinforcement learning (RL), or the empirical distribution (ED).

Table 2 presents the dynamic risk obtained from training the DRM policy for a one year maturity option and applying it on the test data for maturity ranging from 1 to 12 months. Similar to the vanilla option case, the dynamic risk of the writer is monotonically decreasing as we get closer to the maturity of the option, while for the writer the monotonic behavior seems to be slightly perturbed by estimation error. The table also compares the static risk under DRM and SRM. One can first recognize the same monotone convergence to zero of the two sides of the options. However, contrary to the case of the vanilla option, the difference between the static risk performance of DRM and SRM policies are rather similar for all maturity times. It therefore appears that in these experiments with a basket option, both SRM and DRM produce more similar polices. One possible reason could be that the range of "optimal" risk averse investment plans, whether using DRM or SRM, is more limited. Indeed, while for the vanilla option, we observed that the optimal policies generated investments in the range $[0, 1]$ and $[-1, 0]$ for the writer and the buyer respectively, for the basket option we observed wealth allocations that are more concentrated around 0.20 (i.e. the uniform portfolio known for its risk hedging properties) and -0.20 for each of the 5 assets asset respectively. Finally, Table 2 presents the equal risk prices computed based on our RL based out-of-sample dynamic risk estimator. Once again, the higher ERP price for the SRM policy are notable, which again can be attributed to the better performing (in terms of dynamic risk) hedging policy produced by ACRL for the DRM, compared to the policy produced by AORL for the SRM. Further details are presented in section B.3 of the Appendix.

## 5    CONCLUSION

Motivated by the application of ERP, in this paper we considered solving risk averse MDP problems formulated based on dynamic expectile risk measures, and proposed a novel ACRL algorithm that extends the model-free off-policy deterministic ACRL algorithm to a general finite horizon risk-averse MDP setting. In comparison to existing model-free deep RL methods for solving risk-averse MDP formulated based on dynamic risk measures, our method is more amenable to practical implementation, allowing for tackling real applications such as the ERP problem. Indeed, as a natural risk-averse extension of the popular model-free DDPG, our method can easily accommodate any finite horizon MDP applications solved by DDPG. More in-depth studies of these other applications are left for future work. The extension of our method to an infinite horizon MDP setting is also worth investigating further. Finally, the exploration of our method to accommodate other utility-based shortfall risk measures should also be of great interest for future study.

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
