# OpenReview forum: "Deep Reinforcement Learning for Equal Risk Option Pricing and Hedging under Dynamic Expectile Risk Measures"
_ICLR.cc/2022/Conference — ICLR 2022 Submitted_

### Official Review · Reviewer_VoGN · 2021-11-02

**Correctness:** 3
**Technical Novelty And Significance:** 3
**Empirical Novelty And Significance:** 3
**Recommendation:** 6
**Confidence:** 2

**Main Review:**

The work is novel since, as highlighted by the authors, they provide the first DDPG-like algorithm optimizing a dynamic risk-measure. Differently from previous approaches, their work allow to produce time-consistent hedging policies.
One problem with this article consists with its accessibility to a wide audience: the financial introduction may result difficult to understand to a reader who is unfamiliar to option hedging. The paper has also some other passages which may be difficult to follow, for instance in the derivation of the DDPG-like algorithm and in the experimental analysis.
In particular I would like to ask the authors for two specific issues.
1) A stochastic gradient step is used to improve expectile estimators taken from (Shen et al., 2014): can the authors elaborate more on why the suggested one is a good estimate?
2) In section 4.3 you say that "First, unlike with DRM, the static risk of SRM policies for the seller (resp. buyer) can increase (resp. decrease) when hedging an option with shorter maturity. The possibility that a seller’s policy may actually increase risk when applied to an option with shorter maturity is clearly problematic here", however, it is not clear to me how this is apparent from the results of Table 1.

Moreover, it is not clear to me wheter the algorithm is applicable only to option hedging or it always applicable in MDP where the objective is the dynamic risk-measure in exam.
The article also lacks a conclusion, which the author should surely add to their final version in order to summarize their work and to indicate some directions for future research.

**Summary Of The Paper:**

The paper offers a solution to the problem of Equal Risk Pricing (ERP) by framing it as a risk-sensitive MDP formulation. In particular, the authors considers a dynamic risk-measures using expectiles measures as conditional risk-measure for each step. An extension of DDPG is derived by exploiting the properties of the chosen risk-measure and it is tested on two experiments. The first one involves a vanilla option, while the second one feature a more complex basket option.

**Summary Of The Review:**

The paper offer a novel perspective on Equal Risk Pricing for option hedging using reinforcement learnin and dynamic risk-measures. The contribution is mainly an algorithmical and experimental one, but the proposed approach seems to be sound and effective. I would like to propose a borderline accept to this paper.

---

> ### Author Response · Authors · 2021-11-16
> **Response to reviewer VoGN**
>
> We thank the reviewer for his/her response. The new revision presents our edits in blue.
>
> a) Regarding “One problem with this article consists with its accessibility to a wide audience”: following the reviewer’s comment, we completely rewrote the title, abstract and introduction, and added a conclusion, which should make the paper more accessible to an ML audience. In particular, we added a note at the end of the introduction informing a reader that is mostly interested in the new algorithm that he/she can skip right ahead to section 3.
>
> b) Regarding “the derivation of the DDPG-like algorithm”: we included Proposition 3.1 to help with the presentation of the algorithm.
>
> c) Regarding 1.: We thank the reviewer for his/her suggestion. We now have clarified in what capacity the stochastic gradient step is a "good" one. Specifically, for the tabular MDP setting, Shen et al. (2014) prove in Theorem 3.2 that within a properly designed Q-learning algorithm, the estimator will converge almost surely to the true risk at each $t$, $s_t$, and $a_t$.
>
> d) Regarding 2. “however, it is not clear to me how this is apparent from the results of Table 1”:  The reviewer is right and we thank him/her for his/her attention to details. This statement was actually pointing to Figure 6(e) and (f) in the appendix (i.e. with a risk aversion level is of 95%), where the curves for the SRM policy (in orange) is non-monotonically decreasing. We corrected the text accordingly.
>
> e) Regarding 2. “Moreover, it is not clear to me whether the algorithm is applicable only to option hedging or it is always applicable in MDP where the objective is the dynamic risk-measure in exam”: we confirm that ACRL Algorithm 1 can easily be generalized to other problems or application domains. In fact, the version presented in Algorithm 2 in appendix A.2 (which used to be Appendix A) can straightforwardly be applied to ANY problem or application domain that can be modeled as a finite horizon MDP as long as a dynamic expectile risk of the (possibly discounted) sum of rewards is used as the objective. In the revision, we modified Section 3 to  be more clear about this. In particular, we refer the reviewer to the new Proposition 3.1, which applies to any general finite horizon risk-averse MDP. The proposition only assumes that the “offline policy” and initial state distribution is such that there is a strictly positive probability of reaching all of the states in $\mathcal{S}$. This is a typical assumption made for off-policy RL algorithms and is usually met simply by adding a random noise to the policy used to produce the trajectories. Overall, we claim that our ACRL algorithm can be used anywhere the original DDPG is used.
>
> f) Regarding “The article also lacks a conclusion”: we added a conclusion in the revision.

---

> > ### Comment · Reviewer_VoGN · 2021-11-29
> > **Answer**
> >
> > Thank you for your feedback, I think the changes you made improved the quality of your paper. I will keep my positive mark.

---

### Official Review · Reviewer_wndf · 2021-11-07

**Correctness:** 3
**Technical Novelty And Significance:** 3
**Empirical Novelty And Significance:** 3
**Recommendation:** 6
**Confidence:** 4

**Main Review:**

Strengths:
1. The target problem is interesting and maybe potentially useful.
2. The proposed algorithm seems to be rigorous, the theoretical guarantees are appreciated.

Weaknesses:
1. A first question I would hope the authors to clarify is that: from the organization and presentation of this paper, this paper is more like an application paper of DRL to pricing and hedging tasks. But for an AI conference, the readers are expecting to see some general form behind, so that this solution can also benefit related studies. If it focuses on one particular application, then an ML&finance conference or journal would be a better choice for this work.
2. From Alg. 1, one key change from DDPG to the proposed ACRL is taking an average of N, which would effectively reduce variance.  Then, I believe it would be more valuable to discuss the variance.
3. As for the experiment results in Section 4.3 and Section 4.4, the major issue is that the rationale of performance metrics are not clearly presented, so it is not easy to evaluate the results.

After rebuttal:
1. The first concern is addressed.
2. The second one still remains, so there is a question about novelty.
3. The third one still remained, which is still not very easy to evaluate its values to practioners.

**Summary Of The Paper:**

This paper targets a fundamentally important issue: risk-aversion in pricing and hedging. The authors claims the first algorithm to identify optimal risk averse for option hedging strategies, which are time-consistent with respect to a dynamic convex riskmeasure, etc., those are impressive.

**Summary Of The Review:**

Interesting application, but need the authors to justify some ambiguity.

---

> ### Author Response · Authors · 2021-11-16
> **Response to reviewer wndf**
>
> We thank the reviewer for his/her response. The new revision presents our edits in blue.
>
> a) Regarding1.: We appreciate the reviewer’s comment and decided to revise the title, abstract, introduction, and part of Section 3 in order to put more emphasis on the contribution and appeal to ML (namely risk averse RL). In particular, we now present Proposition 3.1 that presents the basic assumptions on the MDP for our ACRL algorithms (either Algorithm 1 in the case of a simulation based environment or Algorithm 2 otherwise) to be employable. These are similar assumptions to those made for other off-policy RL algorithms, namely that the “offline policy” and initial state distribution be such that there is a strictly positive probability of reaching all of the states in $\mathcal{S}$.  This assumption is usually met simply by adding a random noise to the policy used to produce the trajectories. To clarify that our method is not limited to the ERP application, we now added a conclusion section where we point out that as a natural risk-averse extension of the popular model-free DDPG, our method can easily accommodate any finite horizon MDP applications solved by DDPG. Overall, we claim that our ACRL algorithm can be used anywhere the original DDPG is used thus should definitely “benefit related studies”.
>
> b) Regarding 2.: we respectfully disagree with the reviewer here (unless we misunderstood his/her comment). Specifically, we believe that the main difference between the original DDPG and our extension is a change in the loss function used to update the main critic network. Noticing that this might have not been very clear in the original submission, our revision now highlights this step in bold in Algorithm 1 and mentions exactly the distinction when introducing Algorithm 2 in Appendix A.2. Regarding “one key change from DDPG to the proposed ACRL is taking an average of N”, we should clarify that the averaging of the gradient that we do over the elements of the minibatch also appears implicitly in the original DDPG algorithm in the form of “Update critic by minimizing the loss: $L=(1/N)\sum_i (y_i-Q(s_i,a_i|\theta^Q))^2$" where a mean square error is measured over the whole minibatch.
>
> c) Regarding 3.: We added a sentence to the description of the three metrics to clarify the rationale.

---

### Official Review · Reviewer_2QwB · 2021-11-08

**Correctness:** 3
**Technical Novelty And Significance:** 2
**Empirical Novelty And Significance:** 2
**Recommendation:** 5
**Confidence:** 4

**Main Review:**

Overall, the paper is well-written but certain parts, especially finance-related concepts could be further elaborated for the general AI/ML audience.

The paper addresses the important issue of risk-sensitive RL. However, the focus here is very narrow, on applications in option hedging and equal-risk pricing. Usually, when it comes to methods targeting a specific application, the paper must at least achieve one of the following two:
1) Can the proposed approach be easily applied or generalized to other problems or application domains ?
This is not obvious. The dynamic risk measures have very good theoretical properties -- everything is Markov, and therefore very convenient for dynamic-programming-based solutions. The proposed DDPG extensions are therefore quite straightforward compared to the case of static risk measures. Furthermore, as pointed out by the authors, the state-transition dynamic depends only on external uncertainties thereby making the problem even easier than solving general MDPs.

2) If the above condition is not satisfied, then the target application must itself be significant (e.g. solve chess).
Here, the problem is option hedging and equal-risk pricing. I can accept these as significant but the evidence presented in the paper is insufficient.
 - Firstly, all experiments are only done in simulated environments based on the fitted Brownian-motion model (for both training and testing). This means that RL can be trained using unlimited data and there is no train-test mismatch. This does not test the RL ability to generalize beyond the training examples. We have many years of stock prices available, so why not test on real data?
 - The comparison between policies obtained via optimizing static/dynamic risks seems confusing. It seems that using methods that optimize the static risk directly, we would obtain a better test-set static risk. Similarly, using methods optimizing the dynamic risk directly, we would obtain better test-set dynamic risk. What is wrong here?



**Summary Of The Paper:**

The authors propose a risk-sensitive reinforcement learning algorithm based on DDPG for the problem of risk-averse option hedging and equal risk pricing based on dynamic expectiles. The proposed algorithm is empirically tested on a simulated option hedging environment.

**Summary Of The Review:**

Based on my concerns above, I cannot recommend acceptance at this stage.

---

> ### Author Response · Authors · 2021-11-16
> **Response to reviewer 2QwB**
>
> We thank the reviewer for his/her comments. The new revision presents our edits in blue.
>
> a) Regarding “However, the focus here is very narrow, on applications in ...”: we respectfully disagree with the reviewer here as our claim is that the paper has both strong methodological and practical contributions. Methodologically, it is the first to extend the DDPG algorithm to a time-consistent risk averse setting where a coherent risk measure is used. Practically, it is the first to develop a model-free algorithm capable of identifying optimal risk averse option hedging strategies that are time-consistent with respect to a coherent risk measure in an environment rich enough to model basket options. In this revision, we revised the title, abstract and introduction, and presented the two contributions in this new order (methodological before practical) in order to emphasize our methodological contribution. We also added a conclusion section where we point out that as a natural risk-averse extension of the popular model-free DDPG, our method can easily accommodate any finite horizon MDP applications solved by DDPG.
>
> b) Regarding 1: we confirm that our proposed ACRL can easily be generalized to other problems or application domains. In fact, the DDPG extension presented in Algorithm 2 in Appendix A.2 (which used to be Appendix A) can straightforwardly be applied to ANY problem or application domain that can be modeled as a finite horizon MDP as long as a dynamic expectile risk of the (possibly discounted) sum of rewards is used as the objective. In the revision, we modified Section 3 to  be more clear about it. In particular, we refer the reviewer to the new Proposition 3.1, which applies to any general finite horizon risk-averse MDP. The Proposition does not assume that “the state-transition dynamic depends only on external uncertainties” but rather that the “offline policy” and initial state distribution be such that there is a strictly positive probability of reaching all of the states in $\mathcal{S}$. This is a typical assumption made for off-policy RL algorithms and is usually met simply by adding a random noise to the policy used to produce the trajectories. Overall, we claim that our ACRL algorithm can be used anywhere the original DDPG is used.
>
> c) Regarding 2, 1st bullet: We first wish to bring to the reviewer's attention that for financial applications such as the ERP problem, there is a necessity to benchmark in environments with known stochastic dynamics, which is well-known to already constitute a challenge. Indeed, financial institutions heavily employ well calibrated stochastic models to make pricing decisions in order to avoid overfitting the data, facilitate backtesting, and enhance interpretability of the results. This is why we consider it important to start our investigation with the objective of being the first to identify equal risk prices using a dynamic coherent risk measure for a basket option in a setting that employs a geometric Brownian motion. While we see our environment as the first environment in which any DRL approaches for ERP should be tested, we do agree with the reviewer that employing our ACRL algorithm using real data is a natural next step.
>
> d) Regarding 2. 2nd bullet: The reviewer is entirely right in saying that “using methods that optimize the static risk directly, we would obtain a better test-set static risk” and reversely. This indeed shows up in our results. Namely, focusing on the writer problem for the vanilla option, the policy obtained with a static risk measure (to price an option at maturity T=12 months) achieves a static risk of 0.55 compared to 0.56 from the policy of our dynamic expectile risk model (see numbers highlighted in yellow in Table 1). The same occurs for the buyer problem and for both parties in Table 2. In the text, we however make the important (perhaps surprising) observation that the dynamic risk measure policy can outperform the static risk measure policy out-of-sample with respect to static risk if the option that is hedged by the policy has a shorter maturity than 12 months (see numbers highlighted in green). This is a direct consequence of SRM policy not being time consistent. This has two implications : 1) one needs to retrain the SRM policy if one needs to hedge a shorter maturity option even though the policy trained on 12 months does have an action prescribed for the shorter maturity; 2) more importantly after selling the option and while hedging it using the trained policy, if later on the seller also sells a second option, the retrained SRM approach with shorter maturity will dictate a different hedging strategy for this second option even though it has the same maturity (simply because the two options were sold at different time). In 2), the strategy used to  hedge the first option can appear significantly sub-optimal at that point of time. This is an important issue (solved using DRM) from a financial engineering point of view.

---

> > ### Comment · Reviewer_2QwB · 2021-11-21
> > **Re: Response to reviewer 2QwB**
> >
> > Thanks for the response. I appreciate the effort preparing the revised version, which is an improvement.
> >
> > For a methodological contribution, there is little evidence, either analytical or empirical, beyond the ERP applications presented in the paper. This is further diminished by the fact that it is only demonstrated on synthetic data. Usually, to demonstrate the applicability of the proposed method to other domains, at least one different application domain should be added as empirical evidence unless the theoretical contribution is itself significant. In this regard, my feeling is that the paper falls short.
> >
> > That said, I will not complain if the paper is accepted as the other reviewers may have more positive opinions.

---

> > > ### Author Response · Authors · 2021-11-22
> > > **Re: Re: Response to reviewer 2QwB**
> > >
> > >
> > > We thank the reviewer for taking the time to process our response and providing us with timely feedback.
> > >
> > > We wish to point out that the methodological novelty of our work is to show for the first time how DDPG can be analytically generalized to a risk-averse setting that employs dynamic expectile risk measures (remember that the risk level $\tau=0$ recuperates the original DDPG) for a general finite horizon MDP environment. Indeed, prior to our work DDPG could only be applied to the risk neutral or expected utility setting (and not to static or dynamic risk measures). Our work also has a strong contribution for introducing the equal risk pricing problem with dynamic coherent risk measures to the reinforcement learning community and providing additional evidence that DRL can effectively address real financial engineering problems.
> > >
> > > In terms of experimenting with more applications, we must highlight that DRL with coherent risk measures is currently highly under-studied on the empirical side. Tamar et al. (2015) considered a portfolio optimization with three independent assets that follow simple distributions with parameters that were arbitrarily picked by the authors. Huang et al. (2021) considered a tabular environment involving 12 states and 4 actions. In contrast, our test environment is a realistic one involving option hedging and 5 correlated assets, which parameters were calibrated on real market data. The results also strongly showcase the value of using a time-consistent risk averse paradigm, which we believe can provide valuable insights to the reader. While we agree with the reviewer that it would be interesting to test the method in other application domains, we feel that it goes beyond the scope of this conference paper given that our current experimental results already provide a strong basis for considering the usefulness of our new ACRL algorithm.

---

### Comment · Area_Chair_cdwN · 2021-11-20
**Please read responses from the authors**

Dear reviewers,

Please read the detailed responses from the authors. How do they change your evaluation? Do you still have major concerns? Thank you.

---

### Decision · Program_Chairs · 2022-01-20

**Decision:**

Reject

**Comment:**

This paper proposes a risk-sensitive actor critic reinforcement learning (RL) method that optimizes the policy with respect to a dynamic (iterated) expectile risk measure.  The expectile risk measure has the elicitability property and can be expressed as the minimizer of an expected scoring function, which is exploited in critic update.  The proposed approach is applied to option pricing and hedging.

A main point of discussion was the applicability and effectiveness of the proposed method beyond particular financial problems.  The original submission was indeed specialized to particular financial tasks.  The authors have rewritten the paper in a way that it claims to propose a risk-sensitive RL method for the general MDP with finite horizon.  This however leaves the question regarding the advantages of the proposed approach over existing methods of risk-sensitive RL, including those that work with non-coherent (dynamic) risk measures (since coherence is often not needed in domains outside finance).